

# Digital eye strain syndrome among higher education health sciences students in Saudi Arabia: severity and preventive ergonomic practices

Olfat Abdulgafoor Gushgari[1], Samiha Hamdi Sayed[2,3] and
Wafaa Taha Elgzar[4,5]

[1] Public Health Department, College of Health Sciences, Saudi Electronic University, Jeddah, Saudi Arabia
[2] Public Health Department, College of Health Sciences, Saudi Electronic University, Dammam, Saudi Arabia
[3] Community Health Nursing Department, Faculty of Nursing, Damanhour University, Damanhour, Egypt
[4] Department of Maternity and Childhood Nursing, Nursing College, Najran University, Najran, Saudi Arabia
[5] Department of Obstetrics and Gynecology Nursing, Faculty of Nursing, Damanhour University, Damanhour, Egypt

Corresponding author
Samiha Hamdi Sayed,
s.ramadan@seu.edu.sa

## ABSTRACT

**Background:** The increased utilization of digital screens is an unavoidable consequence of the technology era. Digital eye strain (DES) is a prevalent health problem among higher education students in Saudi Arabia, especially health sciences students due to the frequent use of digital sources and virtual classes. Thus, this study aimed to assess the severity of digital eye strain syndrome (DESS) symptoms and preventive ergonomic practices among higher education health science students in Saudi Arabia.

**Methods:** A cross-sectional study using multistage cluster sampling in three cities (Dammam, Riyadh, Jeddah) in Saudi Arabia. A convenient sample of 328 health science students was selected using an equal allocation technique. The researchers designed and used an online survey of three sections: personal and digital device use-related data, the DESS questionnaire, and the self-reported preventive ergonomic practices scale.

**Results:** DESS is a widespread problem among 72.0% of health science students, and 58.2% had unsatisfactory ergonomics. The frequently reported symptoms were blurred vision (32.9%) and increased sensitivity to light (33.5%). The severe eye-related symptoms were headache (45.85%), foreign body inside the eye (43.71%), eye burning (40.19%), and dryness (39.76%). Logistic regression analysis revealed that the female sex, years of screen utilization and the number of hours per day, screen use without rest, and frequency of virtual classes per week, eye disease, digital screen utilization for studying, nonuse of protective eye measures, and using numerous digital devices were significant predictors of DESS risk ($P < 0.05$).

**Conclusions:** DESS is a common problem among many health science students, with headaches and foreign body sensations in the eye being the most frequent symptoms. Screen utilization time and eye disease are significant predictors of DESS, while most

items' application of ergonomic practices was low. Educational programs are needed to increase student's awareness of ergonomic practices.

# INTRODUCTION

In the last decade, the use of technology has increased significantly in all aspects of daily life. Devices such as mobile phones, tablets, desktops, laptops, and various networks are rapidly developing, making the technology widely accessible. Technology is extensively used in health (*Bhavnani, Narula & Sengupta, 2016*; *Golinelli et al., 2020*; *Su et al., 2021*), administration, recreation, and education (*Hubbard Winkler et al., 2011*; *Pedersen, Schneider & Scheckelhoff, 2012*; *Anderson, Subrahmanyam & Cognitive Impacts of Digital Media Workgroup, 2017*). In education, technology offers numerous advantages; it enhances student performance, satisfaction, and engagement (*Vavasseur et al., 2020*). It is also reported to be cost-effective, time-efficient, more accessible, flexible, and supportive of self-learning and frequent training. This is particularly important during the pandemic era, which has significantly increased the rate of e-learning (*Shang & Liu, 2018*).

However, long-term screen use for prolonged periods has numerous psychological, behavioral, social, and health consequences. Health consequences include changes in brain activities, sleep disturbance (*Naeem, 2014*), depression, difficulty in cognitive-emotion regulation, impulsivity, impaired cognition, low self-esteem, reduced physical fitness, eating disorders, decreased cognitive function, headache, and digital eye strain syndrome (DESS) (*Wacks & Weinstein, 2021*). According to the American Optometric Association (AOA), DESS, also known as computer vision syndrome or ocular asthenopia secondary to digital devices, is a group of ocular and vision-related symptoms due to extended use of digital screens, and the symptoms' severity increases as the screen utilization rises (*American Optometric Association, 2023*). The occurrence of digital eye strain (DES) can significantly impact the achievement of three out of the 17 United Nations Sustainable Development Goals (SDGs) goals which aimed for a healthy planet by 2030. In detail, it can significantly affect good health and well-being, quality education, decent work, and economic growth (*Victor et al., 2023*).

Symptoms of DESS can be classified as ocular surface-related symptoms, accommodation or vergence-related symptoms, and extraocular symptoms. Ocular surface-related symptoms caused by decreased eye blinking include eyestrain, tired eye, itching, redness, irritation, eye dryness, foreign body sensation inside the eye, light intolerance, headaches, and discomfort. Accommodation or vergence-related symptoms are related to excessive accommodation and work placed in the binocular system, which incorporates blurred or double vision and difficulty focusing after computer use. Extraocular symptoms are mainly musculoskeletal symptoms that radiate neck and shoulder pain (*Bahkir & Grandee, 2020*; *Kaur et al., 2022*). In the long term, DESS can lead to serious eye problems, affecting students' attention and productivity, increasing error

rates, and worsening visual ability. This can further influence their academic performance and achievement. Furthermore, DESS is a mounting public health problem associated with diminished productivity and overall quality of life, and it is mainly found to be accompanied by poor application of ergonomic principles. Although DESS symptoms are transient and short-lasting, they may be linked to myopia (*Mohan et al., 2022*).

The self-reported prevalence of DESS is 60.0% among males and higher (69.0%) among females, with a mean prevalence of 65.0%, according to the digital eye strain report by *The Vision Council (2016)*. In addition, 73.0% of adults below 30 complained about DESS (*The Vision Council, 2016*). A few studies concerning DESS were conducted in Saudi Arabia despite its rising trend, specifically among high-risk groups such as higher education students during the digitalization era. Two cross-sectional studies reported around 90.0% of the students reported DESS with improper ergonomic practices (*Abudawood, Ashi & Almarzouki, 2020*; *Altalhi et al., 2020*). Different rates were reported for Al Qassim University students (72.0%) (*Al Rashidi & Alhumaidan, 2017*) and King Saud University students (66.0%) in Riyadh (*Al Tawil et al., 2020*). An educational study at the eye clinic of Qassim University, which benefited from the 20/20/20 rule revealed a significant effect on the experience of DESS symptoms. This highlights the pressing need for the present study to assess the students' knowledge about DESS and its preventive ergonomic practices and explore its associated factors. This further can aid in developing need-based educational and prevention programs to enhance students' utilization of preventive ergonomic practices (*Alghamdi & Alrasheed, 2020*).

DESS pathology is complex and multifactorial, and several contributing factors may play a significant role. It may be caused by inadequate lighting, excessive screen glare, and reflection, inadequate contrast between letters and screen background, inappropriate viewing distance, inappropriate body posture, previous untreated vision problems, and sporadic eye blinking during screen utilization (*American Optometric Association, 2023*). In addition, the focus required during screen utilization requires an interaction between the ocular accommodation and convergence mechanisms, which puts people with reflection errors at a higher risk (*Németh et al., 2021*).

Due to the massive utilization of digital devices among all age groups, managing DESS becomes challenging due to the variety of reported symptoms. It is associated with excessive screen utilization, such as educational platforms, distance work or educational activities, and entertainment (video games, watching television or YouTube, and using smartphones for social media), which all correlate to a sedentary lifestyle with severe consequences for eye and general health (*Alamri et al., 2022*). Clear vision on a digital screen requires focusing and refocusing. Frequent eye movement to follow the pixilated pictures and letters can lead to eye strain and fatigue, especially if work continues for more than 4 h. In addition, the short distance between the digital screen and the eye and the prolonged duration of screen utilization is another modifiable risk factor that can lead to aggravation of DESS and myopia (*Holden et al., 2016*).

Simple ergonomics practices can significantly manage DESS. Ergonomics is a scientific discipline that spotlights how the working area can be constructed to match the individual needs instead of individuals' adaptation to the environmental design, which forces them to

be in an uncomfortable and stressful area. Thus, it emphasizes matching individuals' posture and working area to inhibit concurrent musculoskeletal disorders and visual health problems. The AOA highlighted that DESS is an inventible outcome of the protracted use of computers or visual display terminals. It is mainly exaggerated by incorrectly applying ergonomic principles due to a lack of knowledge about its applications (*Center for Disease Control and Prevention, 2020*; *American Optometric Association, 2023*; *World Health Organization, 2024*).

Ergonomics and preventive measures for DESS include decreasing daily time spent on a digital screen to less than 4 h daily, maintaining not less than 50 cm between eye and screen, and taking frequent breaks following the 20-20-20 rule (every 20 min of screen time, take a 20-s break and focus on something 20 feet or more than half a meter away) in addition to maintaining adequate room light, adjusting screen resolutions, using large letter sizes, appropriate screen resolution, and maintaining a healthy posture (upright or on a table). Also, the height of the screen should be maintained below the height of the eye vision angle (15–20° below the eye level) and frequent blinking to decrease eye dryness. The papers used in writing should be positioned above the keyboard level and below the screen. The screen brightness level should be adjusted to match the room lighting, and a screen resolution of 60–70% was suitable for most screen users. Antiglare screens or glasses are suitable for decreasing the reflection from the digital screen. The font size should be 12 in dark color over a bright page background (*Moore, Wolffsohn & Sheppard, 2021*; *Kaur et al., 2022*; *Alamri et al., 2022*). The majority of higher education institutions worldwide and in Saudi Arabia shifted to blended learning during the post-pandemic era. The health sciences students were also asked to submit homework and assignments and conduct quizzes and exams using a digital screen. Some of them conduct research and attend online classes. Therefore, they may use computers for long periods and complain of DESS without being aware of its preventive behaviors. Thus, the present study will be one of the pioneer studies to assess the severity of DESS symptoms and preventive ergonomic practices among higher education health science students, who are considered a particularly high-risk group due to the nature of their study. This can further help develop targeted educational interventions to reduce this rising silent epidemic of DESS and improve students' academic achievement and productivity.

## MATERIALS AND METHODS

### Study design and participants

A cross-sectional design was utilized. The sampling process was done based on three sequential steps. Step 1: Three cities (Dammam, Jeddah, and Riyadh) were randomly chosen. Step 2: One university campus was randomly selected from each city, and the available health college was included in the study. Step 3: A convenient sample of health science students was selected from each college. The inclusion criteria were the students enrolled in health science students, using digital devices for 1 year or more, using digital devices regularly for 4 h or more per day, and being enthusiastic about participating in the study.

## Sampling

The required sample was selected using several parameters, as in the following formula: Z1-α/2 is the standard normal variate for alpha 0.05 (1.96), P is the proportion of DESS in KSA as reported by *Al Rashidi & Alhumaidan (2017)* (70.0%), and d is the precision (0.05). These parameters resulted in a minimum required sample size of 323, and 10% was added to compensate for the expected unresponsiveness rate.

$$\frac{Z_{1-\alpha/2}{}^2 p(1-p)}{d^2}$$

## Data collection

The researchers designed an online survey after thoroughly reviewing relevant recent literature (*Seguí Mdel et al., 2015*; *Assefa et al., 2017*; *American Optometric Association, 2023*). It comprised three parts:

### Part I: personal and digital device use-related data

Age, gender, academic level, co-morbidity, eye problems, wearing medical glasses, eye surgery, previous ophthalmologist consultation, and medication use. In addition to the frequently used digital device (computer, laptop, tablet, smartphone), duration of use/years, daily use/hours, and use of eyeglasses or contact lenses during digital device use, the use of digital learning for study or class (*Seguí Mdel et al., 2015*; *American Optometric Association, 2023*).

### Part II: DES Questionnaire (DES-Q)

It is a self-reported measure of the DESS frequency and intensity of symptoms. It was designed and validated by *Seguí Mdel et al. (2015)* with acceptable reliability (r = 0.78). It comprises 16 items divided by domains: vision-related symptoms (6) and eye-related symptoms (10). The frequency of symptoms in the previous month was rated on a three-point Likert scale: (0) never, (1) occasionally, and (2) frequently. The symptom's severity was scored as 1 = moderate and 2 = severe. The symptom frequency and severity were multiplied for every item. The frequency of each symptom severity was re-coded as (0 = 0) (1 or 2 = 1) or (4 = 2). The student is considered to suffer from DESS if the total score is ≥6 (*Seguí Mdel et al., 2015*).

### Part III: self-reported preventive ergonomic practices scale

It was designed based on the AOA guidelines for DESS prevention. It comprises 17 items/8 dimensions: location of the digital screen (two items), display settings (three items), lighting (three items), antiglare screen/glasses (two items), seating position (three items), document holder (one item), rest breaks (two items), and blinking (one item). It was rated on a five-point Likert scale: always (5) to never (1). The total score ranged from 17 to 85 and was categorized into two levels: unsatisfactory if the total score <60% (<51) and satisfactory if the score was ≥60% (51 and more) (*American Optometric Association, 2023*; *Assefa et al., 2017*).

## Tool validity and reliability

The researchers developed the questionnaire and translated it into Arabic by the DeepL Translator software (DeepL SE Co., Cologne, NW, Germany). It was back-translated by another investigator to confirm its accuracy. A jury of six experts in the field examined its content validity using the Content Validity Index with a satisfactory result (CVI > 70%). The proposed modifications were conducted accordingly. The tool was piloted on 10% of the designated sample size (omitted from the main sample) to appraise its reliability, detect vagueness, and reveal the time elapsed. The Cronbach's alpha coefficient was used to assess the tool's reliability and proved the homogeneity of the items with a statistically acceptable score of its internal consistency (Part II; $\alpha = 0.77$) and (Part III; $\alpha = 0.79$).

## Data collection procedures

The researchers designed the questionnaire using Microsoft Office 365 forms. After ethical and university approval, it was disseminated through Blackboard and students' official emails. It was also disseminated through social media sites such as WhatsApp, Snapchat, Twitter, and Facebook. Students were also encouraged to pass the questionnaire on to their friends. The data collection was conducted over 3 months. The estimated filling time of the questionnaire was 14 min with a high response rate (95.0%). To guarantee the students' correspondence with the preset eligibility criteria, filtered questions were placed at the start of the questionnaire.

## Ethical considerations

The ethical approval was obtained from the institutional review board of the Saudi Electronic University (SEUREC-4505). Ethical consideration was maintained by obtaining digital informed consent from the study subjects and ensuring the anonymity of the collected data. All data was confidential and used for research purposes only. Moreover, the students were informed about their right to withdraw from the study at any time. Informed digital consent was written at the beginning of the questionnaire and made required to proceed with it.

## Statistical analysis

It was conducted using IBM version 23. Number and percentage were used to describe categorical data, while means and standard deviation were used for continuous variables. The total DESS was obtained by multiplying each symptom's frequency and severity, and then the results were summed up for each symptom. The result of the symptom severity score was re-cored as (0 = 0) (1 or 2 = 1) or (4 = 2). If the total DESS score is ≥6, the student is considered to suffer from DESS. Binary logistic regression analysis was utilized to determine the predictors associated with high DESS. The overall fit of the model was tested using the pseudo-$R^2$ value of the Cox & Snell and Nagelkerke tests. The overall significance of the model was tested by the Omnibus test of model coefficients ($P = 0.032$). The model's goodness of fit was tested using the Hosmer and Lemeshow Test ($P = 0.734$).

Bivariate and multivariate analyses were conducted to explore the predictors of high DESS. To handle the potential confounders, all factors in the bivariate analysis with

$P \leq 0.25$ were incorporated into the multivariate analysis model. The independence of observations was confirmed by the multi-collinearity test using the Variance Inflation Factor (VIF > 10). We also analyzed the coefficients of determination (Pseudo $R^2$) and Adjusted Odds Ratios (AOR) along with the 95% Confidence Interval (CI) of each predictor to detect the significant predictors and quantify their effect. The level of association was considered significant at $P < 0.05$.

## RESULTS

### Demographic characteristics, medical history, and digital screen utilization

The mean age of the study participants is 21.58 ± 2.10 years. It also shows the means of the duration of digital screen use (12.55 ± 5.18 years) and daily screen utilization (8.92 ± 4.12 h), screen utilization without a break (123.69 ± 142.17 min per day), and the frequency of virtual classes per week (2.07 ± 0.51 sessions). In addition, 72.3%, 81.1%, and 87.2% were females and had no medical or surgical history, respectively. Furthermore, 27.4% wear medical glasses, and 18.6% use medications daily. More than one-third (34.5%) complained of myopia and used both digital screens only (41.5) or paper sources and digital screens (36.0%) for studying. Moreover, 40.5% of the study participants used numerous digital devices, and 30.8% used laptops. Most study participants did not use protective glasses (84.8%) or eye lenses (95.1) during digital screen utilization (Table 1).

### Frequency of DESS symptoms

Regarding vision-related symptoms, nearly an equal proportion of the study participants (32.9% and 33.5%) often complain of blurred vision and increased sensitivity to light. Around one quarter often complain of difficulty focusing for near vision, colored halos around objects, and poor vision (24.7%, 26.2%, 27.1%, respectively). Concerning the eye-related symptoms, headache (42.4%) was the highest occurring symptom, followed by eye dryness (35.1%), itching (30.8%%), pain (29.0%), burning (29.0%), heavy eyelid (27.1%), and feeling of a foreign body in the eye (25.3%). In total, 72.0% of the study participants suffered from symptoms of DESS (Table 2).

### Severity of DESS vision-related symptoms

Nearly three-quarters of the study participants complain of moderate difficulty focusing for near vision (76.65%), double vision (73.44%), and blurred vision (71.44%). More than two-thirds complained of moderately impaired vision (69.27%), colored halos around objects (62.91%), and increased sensitivity to light (69.43%). An equal proportion suffered from feelings of poor vision (30.73%) and increased sensitivity to light (30.57%). Moreover, around one-quarter complained of difficulty in focusing (23.35%), double vision (26.56%), and blurred vision (28.77%) (Fig. 1).

### Severity of DESS eye-related symptoms

Regarding the severe form of DESS eye-related symptoms, severe headache (45.85%) is the most reported symptom, followed by a foreign body inside the eye (43071%), eye burning

**Table 1 Demographic characteristics, medical history, and digital screen utilization ($n$ = 328).**

| Variable | M ± SD | | |
|---|---|---|---|
| Age in years (M ± SD) | 21.58 ± 2.10 | | |
| Duration of screen utilization (years) | 12.55 ± 5.18 | | |
| Duration of screen utilization: hours/day) | 8.92 ± 4.12 | | |
| Duration of screen utilization without a break in minutes | 123.69 ± 78.17 | | |
| Frequency of virtual classes per week | 2.07 ± 0.51 | | |
| | **Categories** | **No** | **%** |
| Sex | Male | 91 | 27.7 |
| | Female | 237 | 72.3 |
| Medical history | free medical history | 266 | 81.1 |
| | Diabetes | 24 | 7.3 |
| | Asthma | 10 | 3.0 |
| | Hypothyroidism | 9 | 2.7 |
| | Anemia | 19 | 5.8 |
| History of eye surgery | No | 286 | 87.2 |
| | Yes | 42 | 12.8 |
| Wearing medical glasses | No | 228 | 69.5 |
| | less than 1 year | 10 | 3.0 |
| | 1 year and more | 90 | 27.4 |
| Using medication daily | No | 267 | 81.4 |
| | Yes | 61 | 18.6 |
| Eye related problem | Non | 200 | 61.0 |
| | Myopia | 113 | 34.5 |
| | Hyperopia | 7 | 2.1 |
| | Astigmatism | 5 | 1.5 |
| | eye dryness | 3 | 0.9 |
| The most utilized method of studying | Paper study | 74 | 22.6 |
| | Screen utilization | 136 | 41.5 |
| | Both | 118 | 36.0 |
| The most used electronic device daily | Numerous devices | 133 | 40.5 |
| | Mobile | 69 | 21.0 |
| | Laptop | 101 | 30.8 |
| | Tablet | 19 | 5.8 |
| | Desktop | 6 | 1.8 |
| Using protective glasses during screen utilization | No | 278 | 84.8 |
| | Yes | 50 | 15.2 |
| Using protective eye lenses during screen utilization | No | 312 | 95.1 |
| | Yes | 16 | 4.9 |

(40.19%), and eye dryness (39.76%). Around one-third complained of excessive blinking (31.71%), frequent tears (36.59%), eye redness (30.69%), itching (35.92%), heavy eyelids (30.67%), and eye pain (35.82%) (Fig. 2).

**Table 2 Frequency of DESS symptoms (N = 328).**

| | Never | | Occasionally | | Often/always | |
|---|---|---|---|---|---|---|
| | N | % | N | % | N | % |
| **Vision-related symptoms** | | | | | | |
| Blurred vision | 85 | 25.9 | 135 | 41.2 | 108 | 32.9 |
| Double vision | 190 | 57.9 | 85 | 25.9 | 53 | 16.2 |
| Difficulty focusing for near vision | 120 | 36.6 | 127 | 38.7 | 81 | 24.7 |
| Increased sensitivity to light | 73 | 22.3 | 145 | 44.2 | 110 | 33.5 |
| Colored halos around objects | 153 | 46.6 | 89 | 27.1 | 86 | 26.2 |
| The feeling of poor vision | 103 | 31.4 | 136 | 41.5 | 89 | 27.1 |
| **Eye related symptoms** | | | | | | |
| Eye pain | 89 | 27.1 | 144 | 43.9 | 95 | 29.0 |
| Heavy eyelids | 153 | 46.6 | 86 | 26.2 | 89 | 27.1 |
| Burning | 106 | 32.3 | 127 | 38.7 | 95 | 29.0 |
| Itching | 105 | 32.0 | 122 | 37.2 | 101 | 30.8 |
| Eye redness | 139 | 42.4 | 108 | 32.9 | 81 | 24.7 |
| Eye dryness | 75 | 22.9 | 138 | 42.1 | 115 | 35.1 |
| Frequent tear | 151 | 46.0 | 104 | 31.7 | 73 | 22.3 |
| Excessive blinking | 144 | 43.9 | 119 | 36.3 | 65 | 19.8 |
| The feeling of a foreign body | 145 | 44.2 | 100 | 30.5 | 83 | 25.3 |
| Headache | 47 | 14.3 | 142 | 43.3 | 139 | 42.4 |
| **Total reported symptoms of DESS- n (%)** | | | | | | |
| Sufferers | 236 (72.0) | | | | | |
| Non-sufferers | 92 (28.0) | | | | | |

## Preventive ergonomic practices of DESS

Unsatisfactory preventive ergonomic practices during digital screen utilization were detected among 58.2% of the study participants. Regarding the digital screen location, only 19.8% always place the device screen approximately 15–20 cm below eye level from the center of the screen, and 17.7% put it 45–70 cm away from the eye. For the display settings, 37.5% always keep the screen brightness low, 38.1% use dark colors for the screen wallpaper, and 43.6 use the appropriate font size on the screen. Regarding environmental lighting, 31.7% positioned the screen appropriately to avoid glare or harsh light from lamps or windows, 25.3% put curtains on the windows while using a digital screen, and 18.6% used low-light lamps during screen utilization, respectively. Concerning using antiglare devices, 19.8% used a protective screen, 26.85% used glasses or contact lenses while using digital devices, 18.0% took a 15-min break for the eyes every 2 h, and 11% practiced the 20-20-20 rule, 17.7 always practice frequent blinking, and 16.5% use a document holder when typing on the device. Finally, for seating position, 23.5% place their feet in a comfortable position or on the floor while using a digital device, 17.1% adjust the

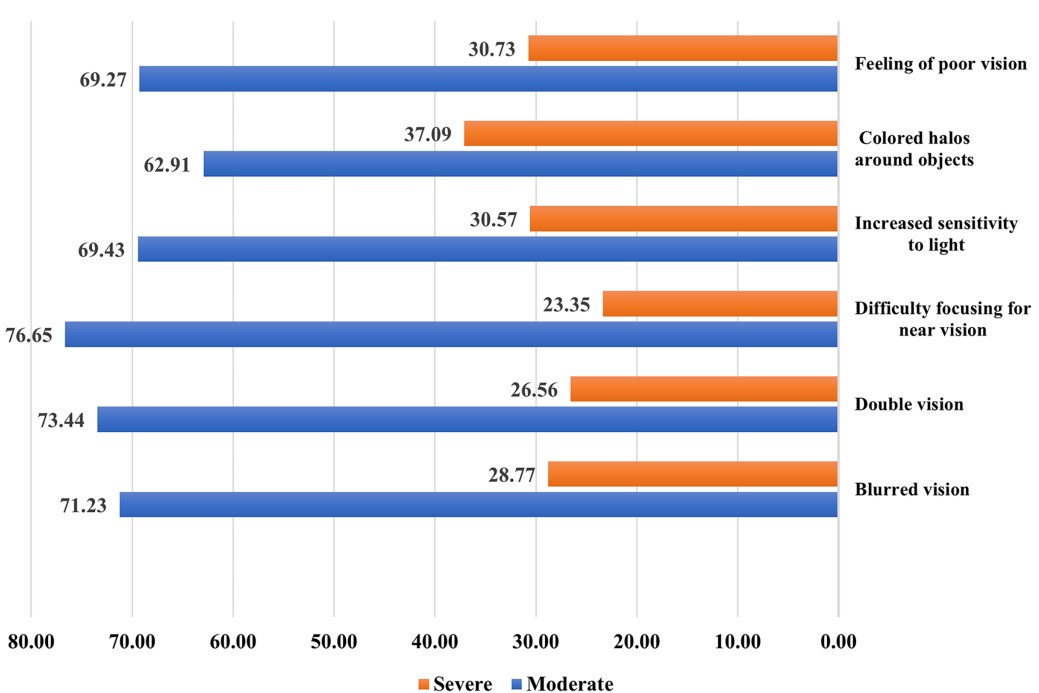

**Figure 1 Severity of DESS vision-related symptoms.**

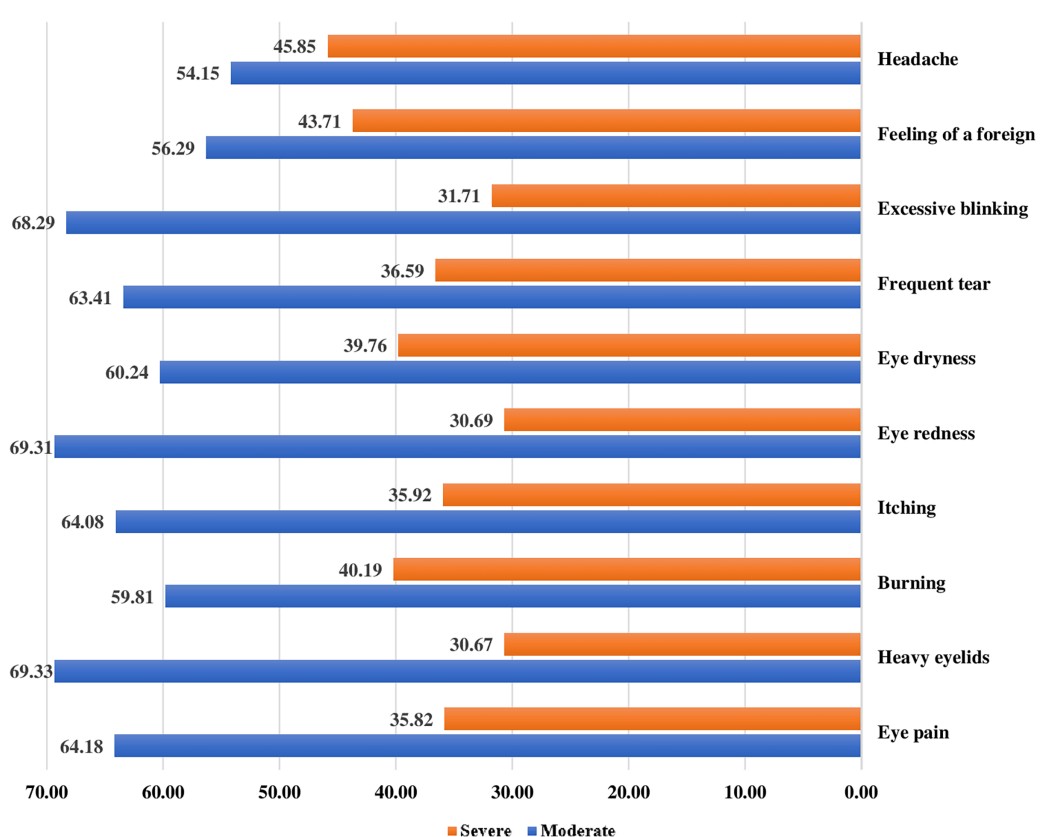

**Figure 2 Severity of DESS eye-related symptoms.**

Table 3 Preventive ergonomic practices of DESS (N = 328).

| | Never | | Rarely | | Sometimes | | Often | | Always | |
|---|---|---|---|---|---|---|---|---|---|---|
| | N | % | N | % | N | % | N | % | N | % |
| **Digital screen location** | | | | | | | | | | |
| Put the device screen approximately 15–20 cm below eye level from the center of the screen. | 46 | 14.0 | 47 | 14.3 | 60 | 18.3 | 110 | 33.5 | 65 | 19.8 |
| Put the device screen 45–70 cm from the eye. | 46 | 14.0 | 47 | 14.3 | 87 | 26.5 | 90 | 27.4 | 58 | 17.7 |
| **Display settings** | | | | | | | | | | |
| Keep the device's screen brightness low. | 19 | 5.8 | 39 | 11.9 | 74 | 22.6 | 73 | 22.3 | 123 | 37.5 |
| Using dark colors for screen wallpaper. | 24 | 7.3 | 39 | 11.9 | 71 | 21.6 | 69 | 21.0 | 125 | 38.1 |
| Use the appropriate font size on the screen. | 22 | 6.7 | 37 | 11.3 | 35 | 10.7 | 91 | 27.7 | 143 | 43.6 |
| **Environmental lightening** | | | | | | | | | | |
| Position the screen appropriately to avoid glare or harsh light from lamps or windows. | 36 | 11.0 | 33 | 10.1 | 58 | 17.7 | 97 | 29.6 | 104 | 31.7 |
| Put curtains on the windows while using a digital screen. | 57 | 17.4 | 57 | 17.4 | 45 | 13.7 | 86 | 26.2 | 83 | 25.3 |
| Using low light lamps (low wattage) during screen use. | 54 | 16.5 | 72 | 22.0 | 53 | 16.2 | 88 | 26.8 | 61 | 18.6 |
| **Antiglare screen or glasses/lenses** | | | | | | | | | | |
| Using a protective screen during screen utilization. | 117 | 35.7 | 48 | 14.6 | 36 | 11.0 | 62 | 18.9 | 65 | 19.8 |
| Using protective glasses or contact lenses while using digital devices. | 119 | 36.3 | 44 | 13.4 | 36 | 11.0 | 41 | 12.5 | 88 | 26.8 |
| **Rest breaks** | | | | | | | | | | |
| Take a 15-min break for the eyes every 2 h during digital device use. | 57 | 17.4 | 58 | 17.7 | 76 | 23.2 | 78 | 23.8 | 59 | 18.0 |
| Practice the 20-20-20 rule. | 108 | 32.9 | 76 | 23.2 | 42 | 12.8 | 66 | 20.1 | 36 | 11.0 |
| **Blinking** | | | | | | | | | | |
| Practice frequent blinking while using the device. | 73 | 22.3 | 83 | 25.3 | 66 | 20.1 | 48 | 14.6 | 58 | 17.7 |
| **Document holder** | | | | | | | | | | |
| Using a document holder when typing on the device or placing the document on the keyboard. | 104 | 31.7 | 77 | 23.5 | 42 | 12.8 | 51 | 15.5 | 54 | 16.5 |
| **Seating position** | | | | | | | | | | |
| Put the feet comfortably or on the floor while using a digital device. | 51 | 15.5 | 65 | 19.8 | 50 | 15.2 | 85 | 25.9 | 77 | 23.5 |
| Adjust the chair's handles to support the arm and wrist while typing on the keyboard. | 88 | 26.8 | 48 | 14.6 | 55 | 16.8 | 81 | 24.7 | 56 | 17.1 |
| Use a comfortable padded chair that conforms to the body. | 68 | 20.7 | 55 | 16.8 | 64 | 19.5 | 67 | 20.4 | 74 | 22.6 |
| **Total preventive ergonomic practices** | *n* | % | | | | | | | | |
| Satisfactory | 137 | 41.8 | | | | | | | | |
| Unsatisfactory | 191 | 58.2 | | | | | | | | |

chair's handles to support the arm and wrist while typing on the keyboard, and 22.6% use a comfortable padded chair that conforms to the body (Table 3).

## Logistic regression analysis for the predictors of high DESS frequency and severity

The current model can predict 40.7% of the risk of DESS. Sex, duration of screen utilization in the year, hours per day, work without rest, and frequency of virtual classes per week were positive predictors for DESS. In detail, being female increased the susceptibility of DESS by 203 times [AOR = 2.31 (1.211–5.230), $P$ = 0.030], each year increase in the duration of screen utilization can nearly double the risk for DESS [AOR = 2.102

**Table 4 Logistic regression analysis of the predictors of high DESS.**

| Variable | AOR | Univariant P value | 95% CI | AOR | Multivariant P value | 95% CI |
|---|---|---|---|---|---|---|
| Age in years | 2.621 | 0.042* | [1.131–5.320] | 1.797 | 0.323 | [0.687–4.700] |
| Duration of screen utilization (years) | 2.239 | 0.001* | [1.058–5.204] | 2.102 | 0.009* | [1.161–5.134] |
| Duration of screen utilization (hours/day) | 1.128 | 0.000* | [1.058–1.204] | 1.171 | 0.000** | [1.071–1.279] |
| Duration of screen utilization without a break in minutes | 1.902 | 0.034* | [1.125–2.720] | 1.372 | 0.043* | [1.231–2.623] |
| Frequency of virtual classes per week | 1.658 | 0.039* | [0.785–3.015] | 1.571 | 0.041* | [0.891–3.123] |
| **Sex** | | | | | | |
| Male | Ref | | | | | |
| Female | 2.430 | 0.001* | [1.451–4.068] | 2.231 | 0.030* | [1.211–5.230] |
| **Medical history** | | 0.023 | | | 0.539 | |
| Free medical history | | Ref | | Ref | | |
| Diabetes | 1.371 | 0.021* | [1.159–1.864] | 0.557 | 0.598 | [0.063–4.898] |
| Asthma | 0.371 | 0.126 | [0.104–1.320] | 0.276 | 0.115 | [0.056–1.366] |
| Hypothyroidism | 2.969 | 0.309 | [0.365–24.58] | 2.044 | 0.582 | [0.160–26.045] |
| Anemia | 3.155 | 0.131 | [0.711–13.996] | 0.683 | 0.741 | [0.071–6.530] |
| **History of eye surgery** | | | | | | |
| No | Ref | | | | | |
| Yes | 1.508 | 0.239 | [0.762–2.986] | | | |
| **Wearing medical glasses** | | 0.007* | | | 0.009* | |
| No | Ref | | | Ref | | |
| Less than 1 year | 4.500 | 0.157 | [0.560–36.174] | 4.721 | 0.238 | [0.714–32.211] |
| 1 year and more | 2.500 | 0.004* | [1.346–4.643] | 2.321 | 0.007* | [1.452–4.322] |
| **Using medication daily** | | | | | | |
| No | Ref | | | | | |
| Yes | 1.011 | 0.972 | [0.544–1.880] | | | |
| **Eye related problem** | | 0.000** | | | | |
| Non | Ref | | | | 0.009* | |
| Myopia | 8.934 | 0.000** | [4.127–19.340] | 7.238 | 0.001* | [2.370–22.111] |
| Hyperopia | 1.005 | 0.704 | [0.425–1.875] | 0.663 | 0.713 | [0.075–5.898] |
| Astigmatism | 3.021 | 0.008* | [1.167–6.247] | 2.521 | 0.032* | [1.075–5.898] |
| eye dryness | 4.361 | 0.020* | [2.121–15.264] | 3.361 | 0.031* | [3.321–12.321] |
| **The primary utilized studying method** | | 0.009* | | | 0.028* | |
| Paper study | Ref | | | Ref | | |
| Screen utilization | 3.920 | 0.002** | [2.132–9.331] | 4.127 | 0.012* | [2.321–8.741] |
| Both | 1.012 | 0.304 | [0.628–1.725] | 1.010 | 0.423 | [0.712–1.811] |
| **The mostly used daily electronic device** | | 0.024* | | | 0.046* | |
| Numerous devices | Ref | | | Ref | | |
| Mobile | 0.396 | 0.005* | [0.208–0.753] | 0.305 | 0.011* | [0.121–0.766] |
| Laptop | 0.549 | 0.048* | [0.303–0.996] | 0.383 | 0.024 | [0.167–0.881] |
| Tablet | 1.358 | 0.645 | [0.369–5.002] | 1.063 | 0.940 | [0.215–5.259] |
| Desktop | .255 | 0.105 | [0.049–1.333] | 0.288 | 0.281 | [0.030–2.773] |

| Table 4 (continued) | | | | | | |
|---|---|---|---|---|---|---|
| Variable | | Univariant | | | Multivariant | |
| | AOR | P value | 95% CI | AOR | P value | 95% CI |
| Using protective glasses | | | | | | |
| Yes | Ref | | | Ref | | |
| No | 11.489 | 0.001* | [2.731–48.329] | 10.724 | 0.020* | [2.814–45.325] |
| Using protective eye lenses | | | | | | |
| Yes | Ref | | | | | |
| No | 0.352 | 0.173 | [0.078–1.582] | | | |

**Note:**
Cox & Snell $R^2$ = 0.283; Nagelkerke $R^2$ = 0.407.
* Statistically significant at 0.05.
** Statistically significant at 0.001.

(1.211–5.230), $P$ = 0.030]. An increased 1 h in screen utilization daily can increase the risk for DESS by 1.17 times [AOR = 1.171 (1.071–1.279), $P$ = 0.000]. The increased duration of screen utilization without rest in 1 min can enhance the risk for DESS by 1.3 times [AOR = 1.372 (1.231–2.623), $P$ = 0.043] (Table 4).

In addition, one virtual class can increase the risk for DESS by 1.5 times [AOR = 1.571 (0.891–3.123), $P$ = 0.041]. Wearing medical glasses [AOR = 2.321 (1.452–4.322), $P$ = 0.007] myopia [AOR = 7.238 (2.370–22.111), $P$ = 0.001], astigmatism [AOR = 2.521 (1.075–5.898), $P$ = 0.032], and eye dryness [AOR = 3.361 (3.321–12.321), $P$ = 0.031] can increase the risk for DESS by 2.3, 7.2, 2.5 and 3.3 times, respectively. Digital screen utilization for studying increased DESS by 34.1 times [AOR = 4.127 (2.321–8.741), $P$ = 0.012] compared to paper studying. Avoiding protective glasses during screen utilization increased DESS by 10.7 times [AOR = 10.724 (2.814–45.325), $P$ = 0.020]. Lastly, using mobile only [AOR = 0.305 (0.121–0.766), $P$ = 0.011] significantly decreased the risk for DESS compared to numerous devices (Table 4).

## DISCUSSION

The increased utilization of digital screens is an unavoidable consequence of the technology era. DESS is prevalent among students who frequently use digital sources for studying and attending virtual classes. However, DESS gauging is challenging because of different definitions, different evaluation criteria, and the surge of electronic device utilization over time. The current study shows that 72.0% of the participants complain of DESS. Along the same line, *AlQarni et al. (2023)* investigated DESS and its relation to virtual learning during the COVID-19 pandemic. They reported that DESS was present among 68.53% of their participants. Although *AlQarni et al. (2023)* study was conducted online at the time of the COVID pandemic, it reported comparable or slightly lower DES prevalence than the current study that was conducted after ending the COVID quarantine. In addition, the majority of higher education institutions worldwide and in Saudi Arabia shifted to blended learning during the post-pandemic era. This result highlighted that the drawbacks of digital screen utilization expanded after returning to normal life as digital

device utilization became a normal routine in our daily lives. Consequently, this finding stresses the urgent need for DES preventive programs. In addition, an American survey conducted in 2016 found that 65.0% of American adults suffer from DESS, with higher reported symptoms among females (69%) compared to males (60%) (*The Vision Council, 2016*).

On the other hand, a Saudi study investigating DESS among medical students at King Saud bin Abdulaziz University for Health Sciences was performed by *Altalhi et al. (2020)*. They found that 97.3% reported suffering from at least one DESS symptom. Also, *Bahkir & Grandee (2020)* reported a much higher rate of DESS (95.8%). However, they calculated their overall rate by persons who reported at least one DESS symptom. In contrast, the current study estimated the DESS rate based on the DESS Questionnaire (DES-Q) (*Seguí Mdel et al., 2015*), which considers multiplying symptoms' frequency and severity. The total DESS Questionnaire calculation depends on numerous statistical steps, not only the presence or absence of some symptoms.

Severe headache is the most reported symptom among current study participants, followed by the feeling of a foreign body inside the eye, then eye burning and dryness. In addition, around one-third complained of excessive blinking, frequent tears, eye redness, itching, heavy eyelids, and eye pain. Also, nearly three-quarters complain of moderate difficulty focusing for near vision, double vision, and blurred vision. Besides, more than two-thirds complained of moderately poor vision, colored halos around objects, and increased sensitivity to light. In line with the current study results, headache was the most common symptom among participants from numerous previous studies (*Reddy et al., 2013*; *Ranasinghe et al., 2016*; *Altalhi et al., 2020*; *AlQarni et al., 2023*).

*AlQarni et al. (2023)* found that their participants highly reported headaches, eye burning, and eye dryness. In addition, *Altalhi et al. (2020)* also examined DESS among medical students and found that the most common symptoms were headache, poor vision, eye itching, burning, and frequent tears. They added blurred vision, eye redness and dryness, sensitivity to light, eye pain, foreign body sensation, frequent blinking, black halos, double vision, and heavy eyelids as additional symptoms. Especially for headaches, previous studies found a significant relationship between headaches and visual fatigue, small distance between the eye and digital screen, and frequent eye accommodation for a long time without rest (*Ranasinghe et al., 2016*). A similar set of DESS symptoms was also reported among university students in the United Arab Emirates (*Shantakumari et al., 2014*) and medical and engineering university students in Chennai (*Logaraj, Madhupriya & Hegde, 2014*).

Regarding DESS symptoms frequency, our study results indicate that almost all symptoms were present among DESS sufferers in different degrees of frequency and severity. However, other studies reported fewer numbers of DESS symptoms (*Reddy et al., 2013*; *Ranasinghe et al., 2016*; *Altalhi et al., 2020*; *Alghamdi & Alrasheed, 2020*; *AlQarni et al., 2023*). The increased frequency and severity of symptoms may be due to the unique nature of studying in health science colleges, which may increase tiredness and fatigue, contributing to headaches and other DESS symptoms. In addition, most of the teaching activities in the post-pandemic era in higher academic institutions were conducted

virtually (exams, periodic evaluations, seminars, and some lectures), which can extend the screen use time as the leading cause of DESS (*Dahshan & Rosdahl, 2022*).

The current study illustrated that sex, years of screen utilization, daily hours of screen utilization, screen use without rest, and frequency of virtual classes per week were positive predictors for DESS. Wearing medical glasses and having myopia, astigmatism, and eye dryness can significantly increase the risk for DESS. Digital screen use for studying and not using protective glasses can dramatically improve the risk for DESS. On the contrary, using mobile only significantly decreased the risk for DESS compared to numerous devices used. In congruence with the current study results, *Altalhi et al. (2020)* reported that using medical glasses was significantly associated with DESS.

Furthermore, the present study showed that using a digital screen for a long time without a break can significantly increase the participants' risk for DESS. These results can be explained by the fact that eye accommodation is an active process, and a stationary eye position for a long time without enough blinking may lead to fatigue. Frequent blinking, changing vision angle, using antiglare glasses, and looking away from the screen every half an hour can reduce DESS symptoms (*Shantakumari et al., 2014*). Additionally, previous studies reported that long working hours without rest with hand-eye coordination can significantly lead to DESS. Even with frequent rest, working on the computer for more than 8 h daily increases eye strain, dryness, and pain, negatively impacting binocular balance (*Parihar et al., 2016*; *Rossi et al., 2019*; *Auffret et al., 2022*). Another study among 520 office workers in New York City reported that DESS is more prevalent among multiple electronic device users than workers using only one device. They also found that DESS is significantly linked to female workers compared to males (*Portello et al., 2012*). The gender differences in the rate of DESS are interpreted by *Guillon & Maïssa (2010)* that females are mostly more susceptible to dry eye when compared with males.

The current study found that over half of the participants have unsatisfactory preventive ergonomic practices during digital screen utilization. Less than one-fifth always place the screen at an appropriate distance or level and practice comfortable seating. Nearly one-third always keep the recommended brightness and font size. Less than one-third were always concerned about the proper physical environment. Around one-quarter use protective screens, glasses, or contact lenses while using digital devices. In addition, a small proportion took a break and practiced the 20-20-20 rule. Moreover, frequent blinking while using the device and using a document holder when typing on the device was practiced by less than one-fifth. Bright environmental lightning can significantly wash screen images and characters and increase glare and DESS. Therefore, using low room lighting and avoiding screens in front of the window is essential. A previous study reported that computer use in a dark or bright environment can significantly increase DESS (*Shantakumari et al., 2014*), decreasing occupational performance rate by 4–8% (*Harris, Sheedy & Gan, 1992*). Students' reliance on new technology for studying and research can lead to vast hours of work on digital screens, which increases the burden of DESS and can impair students' performance. Thus, it must be addressed and managed as an essential barrier to student productivity and achievement. Therefore, frequent rest periods using a

work-rest schedule are recommended during long working hours, which requires hand-eye coordination (*Toomingas et al., 2014*; *Parihar et al., 2016*).

On the contrary, *Altalhi et al. (2020)* found that 82.0% of the studied medical students adjusted the screen brightness while taking frequent rest periods (66%). They further added that more than half of their participants practiced appropriate ergonomic practices regarding seating, screen position, and antiglare filter use. The differences between *Altalhi et al. (2020)*'s and current results may be related to the scale used to evaluate the participants' adherence to DESS preventive behaviors. *Altalhi et al. (2020)* used yes or no questions to assess ergonomic performance regardless of frequency. The current study evaluates ergonomic performance based on a 5-point Likert scale from never to always, and the researchers considered complete adherence in the case of always performing ergonomic practices. Therefore, the prevalence of ergonomic practices reported by *Altalhi et al. (2020)* may not reflect the real adherence inform of always practicing. The current study highlighted the importance of evaluating ergonomic practices based on the graduation concept which helps to accurately estimate adherence. Thus, universities must make intensive efforts to decrease the contributing factors to DESS among their students by adjusting the class environment and increasing student awareness about appropriate ergonomics practices.

## Study strengths and limitations

The current study tackles an important topic for higher education health sciences students, especially in the recently accommodated e-learning module in the post-pandemic era. DESS was assessed using a validated questionnaire that considered both frequency and severity, not only the presence or absence of symptoms. A 5-point Likert scale is used to assess ergonomic practices, which help to determine the degree of adherence accurately. Numerous human practices are difficult to evaluate with yes or no. Gradualism provides comfort in evaluation as the degree of commitment to ergonomic practices may differ from one person to another. This study respected this gradualism in evaluating commitment to ergonomic practices which provides a more realistic evaluation that respects every trial for adherence. However, some limitations exist, such as bias associated with self-reported questionnaires and the risk of over- or under-estimating symptoms' severity. In addition, the eye problems were identified through self-reported history without an eye examination.

## CONCLUSIONS

The present study concluded that DESS is a widespread problem among health science students in Saudi Arabia, where nearly three-quarters were DESS sufferers, and more than half practiced unsatisfactory ergonomics. Severe headache is the most reported symptom, followed by severe sensation of a foreign body inside the eye, eye burning, and dryness. DESS risk can be predicted by the female sex and increasing years of screen utilization, hours of screen use per day, work without rest, and frequency of virtual class per week. In addition to having a history of eye problems, digital screen utilization for studying, nonuse of protective eye measures, and using numerous digital devices.

### Recommendation and clinical implications

The current study sheds light on the burden of DESS among health sciences students in Saudi Arabia with myriad symptoms and a lack of satisfactory ergonomic practices. Thus, it is advisable to develop targeted educational interventions for this high-risk group to foster their quality of life and improve their academic achievements. Moreover, higher education institutions must incorporate the DESS and its preventive ergonomics in curricula. Policymakers can also develop specific guidelines for regular vision screening of higher education students for early detection of any eye problem to decrease the risk of DESS. Besides, replication of the current study on a larger population and different settings is warranted. Replication of the current study could validate and expand the findings in different contexts and enhance the chance of results generalization.

## ACKNOWLEDGEMENTS

We sincerely thank all the participants for their contributions to this study.

### Funding

This research was supported by the Deanship of Scientific Research at Saudi Electronic University (no. 8267). The funders had no role in study design, data collection and analysis, decision to publish, or preparation of the manuscript.

### Grant Disclosures

The following grant information was disclosed by the authors:
Deanship of Scientific Research at Saudi Electronic University: 8267.

### Competing Interests

The authors declare no competing interests.

### Author Contributions

- Olfat Abdulgafoor Gushgari performed the experiments, authored or reviewed drafts of the article, and approved the final draft.
- Samiha Hamdi Sayed conceived and designed the experiments, analyzed the data, authored or reviewed drafts of the article, and approved the final draft.
- Wafaa Taha Elgzar conceived and designed the experiments, performed the experiments, analyzed the data, prepared figures and/or tables, authored or reviewed drafts of the article, and approved the final draft.

### Human Ethics

The following information was supplied relating to ethical approvals (*i.e.*, approving body and any reference numbers):

Institutional Review Board of the Saudi Electronic University (SEUREC-4457)

## Data Availability

The data and codebook are available in the Supplemental Files.

## Supplemental Information

Supplemental information for this article can be found online at http://dx.doi.org/10.7717/peerj.18423#supplemental-information.

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
