# Peer review of "Digital eye strain syndrome among higher education health sciences students in Saudi Arabia: severity and preventive ergonomic practices"

_PeerJ, doi:10.7717/peerj.18423_

## Round 0.1 · original submission · Minor Revisions

Both reviewers have minor, but valuable comments. Please address all their points.

·

Basic reporting

Good.

Experimental design

Good.

Validity of the findings

Dear authors,
Thanks for your good work. However, I have some concerns that need to be resolved.

Reliability and Validity:
CVI should be more than 0.9 to be a good fit but yours 0.7.

Coronach's alpha less than .8 is just accepted but not good or strong.

You should check the reliability and validity outcomes with your statistian.

Good luck.
Regards,
Prof. Mohammed Iqbal

Additional comments

None.

Reviewer 2 ·

Basic reporting

1. BASIC REPORTING

 CLEAR, UNAMBIGUOUS, PROFESSIONAL ENGLISH LANGUAGE USED THROUGHOUT.:

Overall, the text is clear and professionally written. However, some minor revisions could improve academic clarity and readability.

A sample of corrections is shown for lines 57 to 77 taken from the introduction section (line 66).

Original:
During the last decade, technology utilization increased in an observed manner in all aspects of daily life. Technology terminals such as mobile phones, tablets, desktops, laptops, and different networks are rapidly developing, making technology available everywhere. Technology is widely used in health (Bhavnani et al., 2016; Golinelli et al., 2020; Su et al., 2021), administration, recreation, and education (Hubbard Winkler et al., 2011; Pedersen et al., 2012; Anderson & Subrahmanyam, 2017). Technology utilization in education has numerous and unaccountable advantages; it enhances student performance, satisfaction, and engagement (Vavasseur et al., 2020). It was also reported to be cost-effective, less time-consuming, more accessible, flexible, and enhances self-learning and frequent training. This is of particular concern during the pandemic era, which facilitates the continuation of the learning process, where the rate of e-learning has significantly increased (Shang & Liu, 2018).

Revised for academic clarity and readability:
In the last decade, the use of technology has increased significantly in all aspects of daily life. Devices such as mobile phones, tablets, desktops, laptops, and various networks are rapidly developing, making technology widely accessible. Technology is extensively used in health (Bhavnani et al., 2016; Golinelli et al., 2020; Su et al., 2021), administration, recreation, and education (Hubbard Winkler et al., 2011; Pedersen et al., 2012; Anderson & Subrahmanyam, 2017). In education, technology offers numerous advantages; it enhances student performance, satisfaction, and engagement (Vavasseur et al., 2020). It is also reported to be cost-effective, time-efficient, more accessible, flexible, and supportive of self-learning and frequent training. This is particularly important during the pandemic era, which has significantly increased the rate of e-learning (Shang & Liu, 2018).

The changes made in the revised paragraph are.

• Simplified the opening sentence.
• Removed redundancy.
• Improved flow.
• Clearer enumeration.
• More concise wording.

This type of academic writing will improve the standard of the published article. If the remaining paragraphs are improved with academic clarity and readability, the article will be more standard in the health professional database."


 INTRO & BACKGROUND TO SHOW CONTEXT:
The introduction section should have been linked more explicitly to the aims of the study and the United Nations' Sustainable Development Goals (SDGs). This connection could enhance the representation of national efforts towards achieving the global SDGs within the research context.
Please, refer and include the cited article, which guides how to incorporate the aim of the research work with UNs’ SDGs : “Victor, Virginia M., et al. "A Web-Based Cross-Sectional Survey on Eye Strain and Perceived Stress amid the COVID-19 Online Learning among Medical Science Students." International Medical Education, vol. 2, no. 2, 2023, pp. 83-95, https://doi.org/10.3390/ime2020008. Accessed 27 Jul. 2024”.

 LITERATURE WELL REFERENCED & RELEVANT:

Yes, the literature in the provided content is well-referenced and relevant.


 STRUCTURE CONFORMS TO PEERJ STANDARDS, DISCIPLINE NORMS, OR IMPROVED FOR CLARITY:

No issues found in the structure. The article is structured according to the standards of PeerJ.

 THE FIGURES ARE RELEVANT, HIGH QUALITY, WELL LABELLED & DESCRIBED.

The values on the X-axis do not accurately represent the data in the graphical figure

 RAW DATA SUPPLIED (SEE PEERJ POLICY).
Yes. The raw data was seen in the supplemental docs

Experimental design

2. EXPERIMENTAL DESIGN

 Original primary research within Scope of the journal:

• Yes: the study has followed the steps of the research process and provided the results based on the set objectives and cross-sectional study design. Thus, this article appears to meet PeerJ's requirements for a Research Article in the Health Sciences and would be a suitable submission for their journal.

 Research question well defined, relevant & meaningful. It is stated how the
research fills an identified knowledge gap:

• Yes, the research problem is well-defined, relevant, and meaningful. It addresses an identified knowledge gap by focusing on health science students, who frequently use digital devices for learning. The study strongly suggests practicing ergonomics to prevent Digital Eye Strain Syndrome (DESS) symptoms.

 Rigorous investigation performed to a high technical & ethical standard.
• Clarity is required in the sampling technique: The lines from 166-173 state that a multistage cluster sampling method was utilized, which typically involves random selection at multiple stages. However, it also mentions that a convenient sample of health science students was selected from each college, which contradicts the multistage cluster sampling method. Convenient sampling is non-random and based on availability. While cities were selected randomly, participants were selected conveniently from various colleges in the randomly selected cities. This introduces contradictions and inconsistencies in the sampling methods.
• Ethical considerations have been adhered to according to IRB guidelines.

 Methods described with sufficient detail & information to replicate.
• Yes, the study has provided detailed information about the methodology, used standardized tools, and employed suitable statistical tests to meet the set objectives. The results have been presented with statistical values and significance levels, ensuring transparency and reproducibility. This level of detail allows for validation of the findings and demonstrates that the results are based on rigorous scientific inquiry.

Validity of the findings

3. VALIDITY OF THE FINDINGS

 Impact and novelty are not assessed. Meaningful replication encouraged where
rationale & benefit to literature is clearly stated:

• Validity: The use of a "5-point Likert scale to assess ergonomic practices" (lines 446–447) is highlighted as a novel approach, allowing for a more accurate determination of adherence levels. However, the manuscript could more explicitly state how this contributes new insights compared to previous research methodologies.



Impact and Novelty:
• The manuscript references studies by Al Qarni et al. (2023), Altalhi et al. (2020), and others to compare DESS prevalence and symptoms (lines 345–374). While this situates the study within existing literature, the manuscript does not clearly articulate what sets its findings apart or what new knowledge it adds beyond confirming known issues.

• The manuscript briefly mentions the 'differences between Altalhi et al.’s and current results' regarding ergonomic practices (lines 434–435). However, it misses an opportunity to explore the broader implications of these differences. A deeper analysis could highlight the study's unique contributions to the literature on ergonomic practices and Digital Eye Strain.
Replication:
• The study's robust methodology, including the use of the DESS Questionnaire (DES-Q) (lines 356–357) and a validated ergonomic practices scale, provides a solid foundation for future research, suggesting that replication is warranted. However, the manuscript does not explicitly state why replication would be meaningful or what benefits it might bring to the broader field. It would strengthen its case for replication by outlining how similar studies could validate or expand upon its findings in different contexts.

 All underlying data have been provided; they are robust, statistically sound, &
Controlled:

• The sample size and multistage cluster sampling across major cities indicate efforts to achieve data robustness.
• The use of validated tools for data collection enhances the study's credibility.
• Logistic regression analysis and detailed presentation of results demonstrate a statistically sound approach.
• Consideration of multiple predictors and control for confounding variables strengthen the findings.
• While the cross-sectional design has inherent limitations, the study's approach to data analysis and comparison with other research supports the validity of its conclusions.

 Conclusions are well stated, linked to original research questions & limited to supporting results.
• The conclusions of the manuscript are clear and concise, effectively summarizing the key findings of the study.
• The conclusion clearly identifies severe headaches and foreign body sensations as predominant symptoms, which emphasizes the study's significance and provides a straightforward answer to the research questions (Page 453, Lines 453-455).

• The conclusions are well-linked to the original research questions, which aimed to assess the severity of DESS symptoms and the preventive ergonomic practices among students.
• The statement that "three-quarters of the students experience DESS" directly correlates with the study's aim to determine its prevalence (Page 453, Line 453).
• The conclusions are appropriately limited to the supporting results of the study.
• By focusing on statistically significant findings, such as the role of screen time without rest and lack of ergonomic practices, the authors avoid making unsupported claims (Page 457, Lines 457-459).

Additional comments

Identified research problem is very significant in the current digital era.

---

## Round 0.2 · accepted · Accept

I have now had the opportunity to read your revised manuscript, and your responses to the reviewers' comments. I believe that you have addressed the concerns raised, and I am happy to accept your manuscript.

Reviewer 2 ·

Basic reporting

The suggested amendments have been noted in the resubmitted article.

Therefore, no further comments.

Experimental design

No comments.

Validity of the findings

No comments.

Additional comments

Thank you for adhering to the suggestions. The revisions have greatly improved the clarity and quality of the manuscript, and your efforts are evident in the enhanced final submission.